# Silver-Nanoparticles Embedded Pyridine-Cholesterol Xerogels as Highly Efficient Catalysts for 4-nitrophenol Reduction

**DOI:** 10.3390/ma13071486

**Published:** 2020-03-25

**Authors:** Ganesh Shimoga, Eun-Jae Shin, Sang-Youn Kim

**Affiliations:** Interaction Laboratory of Advanced Technology Research Center, Korea University of Technology and Education, Cheonan-si, Chungcheongnam-do 330-708, Korea; ejshin@koreatech.ac.kr

**Keywords:** cholesterol, heterogeneous catalysis, 4-nitrophenol, silver nanocomposites, xerogels

## Abstract

Two xerogels made of 4-pyridyl cholesterol (PC) and silver-nanocomposites (SNCs) thereof have been studied for their efficient reduction of 4-nitrophenol (4-NP) to 4-aminophenol (4-AP) in the presence of aqueous sodium borohydride. Since *in-situ* silver doping will be effective in ethanol and acetone solvents with a PC gelator, two silver-loaded PC xerogels were prepared and successive SNCs were achieved by using an environmentally benign trisodium citrate dehydrate reducing agent. The formed PC xerogels and their SNCs were comprehensively investigated using different physico-chemical techniques, such as field emission scanning electron microscopy (FE-SEM), Fourier transform infrared (FTIR) spectroscopy, thermogravimetric analysis (TGA), powdered X-ray diffraction (XRD) and UV-Visible spectroscopy (UV-Vis). The FE-SEM results confirm that the shape of xerogel-covered silver nanoparticles (SNPs) are roughly spherical, with an average size in the range of 30–80 nm. Thermal degradation studies were analyzed via the sensitive graphical Broido’s method using a TGA technique. Both SNC-PC (SNC-PC-X1 and SNC-PC-X2) xerogels showed remarkable catalytic performances, with recyclable conversion efficiency of around 82% after the fourth consecutive run. The apparent rate constant (k_app_) of SNC-PC-X1 and SNC-PC-X2 were found to be 6.120 × 10^-3^ sec^-1^ and 3.758 × 10^-3^ sec^-1^, respectively, at an ambient temperature.

## 1. Introduction

A variety of metal-nanocomposites were prepared in recent years, and mostly utilized for advanced catalytic and sensor applications [1,2,3,4,5,6,7,8,9,10,11]. Significant progress has been encountered in the templated synthesis of nanostructured materials with unique structures, morphologies and properties for potential applications in fascinating scientific domains [12,13,14,15,16].

Recently, considerable research interest has been focused on low molecular-mass organic gelators (LMOGs) due to their numerous dynamic applications [17,18,19,20,21]. The entrapping of solvent molecules in three-dimensional self-assembled fibrillar arrangements confine the formation of strong non-covalent interactions between LMOG units and solvents, resulting in network structures signified as gels [22]. Typically, a gel consists of one or more gelators (gelling agents) and a fluid (organic solvent, water or supercritical liquid) which behaves as a soft visco-elastic matter due to the immobilization of solvent molecules in a three-dimensional (3D) network. The gelation ability of any organo-gelators mainly depends upon three important factors: (1) minimum gelation concentration (MGC); (2) phase transition temperature (*T*_gel_); and (3) gelation versatility. Property (1) MGC is the lowest possible concentration of a gelator required to form a stable gel. If the MGC of a gelator is < 1 weight %, such gelators can be denoted as ‘supergelators’. Property (2) mainly focuses on ‘‘thermally-stable galators’’, which have a *T*_gel_ higher than their solvent boiling points. Contrarily, the property (3) of gelators is being studied distinctly by examining the solvent compatibility with gelators. More recently, various LMOGs tripeptoids with glucosamine, cyclohexyl, ethyl glycine and 1-deoxyglucose functional structures were distinctly studied for their gelation versatility [22,23,24,25,26,27,28]. Among various supramolecular gelators, cholesterol-based organogelators are remarkably attractive due to their synthetic tunability and gelation ability [29,30,31,32,33]. Kawano et al. developed pyridine-containing cholesterol compounds, and explored their gelation ability in various organic solvents, claiming that pyridine shows versatile solubility in both protic and aprotic solvents (including water and cyclohexane). They also explained the role of pyridine moiety in gelators and its affinity with solvent molecules. With this background, we have selected to synthesize the versatile pyridine gelator (Figure 1) described previously by Kawano et al. [34].

Since the pyridine has a dipole running from 4-C to 1-N, in a one-dimensional columnar aggregate of PC molecules, the arrangement of dipoles are in radical form around the central cholesterol columns. The helical arrangement of pyridine around the cholesterol column may be responsible for the fine fibril or tubular structure of xerogels. The interactive arrangements of pyridine groups around cholesterol results in complex 3D structures [34,35,36].

Over the past few decades, prime attention has been given to silver-nanocomposites (SNCs) in the design of 3D network templates of SNCs for the efficient reduction of nitroarene compounds, which are widely used in industrial manufacturing units (pharmaceuticals, explosives, dyes, solvents and pigments) and agricultural waste water (pesticides and fungicidal agents). The removal of toxic pollutants from the wastewater is necessary for comestible use. Therefore, the development of nanocatalysts composed of SNPs are in practice to solve this environmental issue [37,38,39,40,41,42]. 

We report herein the successful development of two pyridine-containing silver nanocomposite xerogels, where SNPs are intercalated by the xerogel network. The SNPs-supported xerogels showed excellent catalytic behavior in the reduction of the model reaction of 4-NP to 4-AP. The designed catalysts and its pristine analogue were spectroscopically characterized, and catalytic performances for the reduction of 4-NP to 4-AP in the presence of sodium borohydride were reported. 

## 2. Materials and Methods 

Compound PC is not available commercially; it was synthesized according to previously reported literature [34]. Cholesteryl chloroformate, 4-aminopyridine, trimethylamine, chloroform and silver nitrate (≥ 99.0 %) were purchased from Sigma-Aldrich (Seoul, South Korea), and used without further purification. All the other chemicals and solvents are of reagent grade, and used as received. High performance liquid chromatography (HPLC) grade water was used throughout the study.

### 2.1. Synthesis of 4-pyridyl Cholesterol (PC) Compound 

Compound PC was synthesized from the previously described method [34]. Some alterations are done in the protocol to achieve a better yield. Typically, in a 250 mL round bottom flask, 4-aminopyridine (1.00 g, 10.625 mmol) and trimethylamine (2.36 g, 23.376 mmol) were mixed in 80 mL of chloroform and cooled to 0 °C. To this, a cold (0–5 °C) solution, cholesteryl chloroformate (4.29 g, 9.563 mmol) in 50 mL of chloroform, was added over a period of 15 min. After this addition, the white turbid mixture was allowed to cool to room temperature and stirred for 20 h. The solids in turbid mixture were dissolved by adding an additional 150 mL of chloroform, and after the solids are dissolved, the solution was washed with 200 mL of 0.5 N sodium hydroxide solution. Compound PC was obtained by the precipitation method using base treated chloroform by adding an excess of n-hexane (≈ 500 mL). Solvents were removed using a rotary evaporator to get PC as a white solid. The obtained PC solid was purified by repeatedly dissolving it in chloroform and re-precipitated by adding n-hexane, and this process was repeated at least three times to obtain a pure PC compound (4.8 g, 89.15% yield).

Compound name: PC, white powder; m.p. 192.4 °C [34], (Literature value = 192–193 °C).

Elemental analysis: Anal. calcd. for C_33_H_50_N_2_O_2_ (506.762): C, 78.21; H, 9.94; N, 5.53. Found: C, 78.39; H, 9.78; N, 5.48.

### 2.2. Preparation of 4-pyridyl Cholesterol (PC) Xerogels 

Two xerogels were prepared using the compound PC in solvents ethanol and acetone. Typically, 25 g/dm^3^ of compound PC was dissolved separately in ethanol and acetone. The gel formed after 15 h was allowed to dry at room temperature for two days. The formed xerogels were vacuum dried at 50 °C for 10–12 h. The xerogels formed using PC with ethanol and acetone solvents were designated as PC-X1 and PC-X2, respectively (for pictograph of gel formation, see Appendix A).

Compound name: PC-X1, m.p. 192.1 °C, (Literature value = 192–193 °C).

Elemental analysis: Anal. calcd. for C_33_H_50_N_2_O_2_ (506.762): C, 78.21; H, 9.94; N, 5.53. Found: C, 78.43; H, 9.82; N, 5.37. Final weight of xerogel PC-X1 = 24.91 g (99.64% yield).

Compound name: PC-X2, m.p. 192.7 °C, (Literature value = 192–193 °C).

Elemental analysis: Anal. calcd. for C_33_H_50_N_2_O_2_ (506.762): C, 78.21; H, 9.94; N, 5.53. Found: C, 78.48; H, 9.75; N, 5.32. Final weight of xerogel PC-X2 = 24.88 g (99.52% yield).

### 2.3. Preparation of Silver-Nanocomposites of 4-pyridyl Cholesterol (SNC-PC) Xerogels 

To obtain SNC-PC xerogels, 0.05 M silver nitrate solutions were prepared in ethanol and acetone. 25 g/dm^3^ of compound PC was dissolved separately in (0.05 M) silver nitrate solutions (ethanol and acetone) to obtain Ag-PC-X1 and Ag-PC-X2, respectively. The conditions necessary to obtain silver-loaded xerogels were adopted as described in Section 2.2. The reduction of silver ions was done by trisodium citrate dehydrate. Typically, dried Ag-PC-X1 and Ag-PC-X2 were suspended separately in 50 mL of 1 mM solution of trisodium citrate solution for 1 h with gentle stirring. The obtained silver-nanocomposites of 4-pyridyl cholesterol xerogels (SNC-PC-X1 and SNC-PC-X2) were washed repeatedly with water, and vacuum dried for 10–12 h at 50 °C.

Compound name: SNC-PC-X1.

Elemental analysis: Anal. calcd. for C_33_H_50_N_2_O_2_Ag (506.762): C, 78.21; H, 9.94; N, 5.53. Found: C, 78.37; H, 9.86; N, 5.39. Ag content in xerogel SNC-PC-X1 (Theoretical) = 0.05393 g, Experimental (ICP-OES) 51.2 ppm ± 0.4 (0.8), the relative standard deviation (RSD) values are reported in brackets.

Compound name: SNC-PC-X2.

Elemental analysis: Anal. calcd. for C_33_H_50_N_2_O_2_Ag (506.762): C, 78.21; H, 9.94; N, 5.53. Found: C, 78.42; H, 9.89; N, 5.37. Ag content in xerogel SNC-PC-X2 (Theoretical) = 0.05393 g, Experimental (ICP-OES) 42.8 ppm ± 0.6 (1.2), the RSD values are reported in brackets.

## 3. Characterization Techniques

Since 4-pyridyl cholesterols (PCs) and silver-nanocomposites (SNCs) of xerogels are soluble in tetrahydrofuran (THF), 2-weight % solutions of PC, SNC-PC-X1 and SNC-PC-X2 were prepared in THF, and liquid state UV-Visible spectroscopy (UV-Vis) spectra were recorded on a Perkin-Elmer UV-Vis spectrometer, model UV/VIS-35 (PerkinElmer Inc., Waltham, MA, USA). Thermal stability of PC, PC-X1, PC-X2, SNC-PC-X1 and SNC-PC-X2 were examined using a thermogravimetric analyzer (Q500, TA instruments, New Castle, DE, USA) in the temperature range of between 25 °C to 600 °C in a N_2_ gas flow (50 mL/min) at a 10 °C/min heating rate. The X-ray diffraction (XRD) measurements were done for finely powdered samples of PC, SNC-PC-X1 and SNC-PC-X2, in addition to silver nanoparticles (SNP, Reference material [40]) using Brucker’s D-8 advanced X-ray diffractometer (BRUKER AXS, Inc., Madison, WI, USA). Ni-filtered Cu Kα radiation was used as the X-ray source. The scanning of diffraction was done in the reflection mode from 5 to 90° (2θ) at a constant speed of 8°/min. The Ultra-High-Resolution Field-Emission Scanning Electron Microscope (FESEM, FEI, & Nova NanoSEM450, Thermo Fisher Scientific, Waltham, MA, USA) instrument, operating at 25 kV, was used to analyze the surface morphology of SNCs xerogels. The catalytic performances of SNC-PC-X1 and SNC-PC-X2 were evaluated for the standard conversion reaction of 4-nitrophenol (4-NP) to 4-aminophenol (4-AP) in a standard quartz cell of 10 mL capacity. Typically, 10 mL of aqueous solution of 4-NP (0.1 mM) was mixed with 0.05 g of SNC-PC-X1 or SNC-PC-X2, and later 5 mL aqueous NaBH_4_ solution (50 mM) was added, and the UV-vis absorption spectra was recorded with respect to time using a VARIANEL08043361 UV-vis spectrophotometer (Thermo Fisher Scientific, Waltham, MA, USA). The conversion processes of 4-NP to 4-AP were carefully observed from 250 to 600 nm for 2–3 min intervals at an ambient temperature (22 °C).

## 4. Results and Discussion

### 4.1. UV-Vis Spectroscopic Analysis

The UV-Vis spectrum for 2-weight % solution of 4-pyridyl cholesterol (PC) in THF shows a strong noticeable absorption peak. The most intense absorption, occurring at approximately 190 nm, is related to the π–π* transition for the peptide bond. In the case of SNC-PC-X1 and SNC-PC-X2, however, the lower energy transitions are profoundly dominating, in addition to a characteristic surface plasmon resonance (SPR) peak for SNPs. The lower energy transition, occurring at 265 nm, corresponds to the π–π* transition in the aromatic portion of pyridine moiety (see Figure 2). The inset figure shows a strong SPR peak flanked at 420 nm, due to free-electrons excitation, and their collective oscillations were detected via the Kretschmann configuration [43]. However, it can be noticed that the SPR band is absent in the case of PC.

### 4.2. Functional Group Analysis

Fourier transform infrared (FTIR) spectral analysis has been utilized to verify the functional groups and their interaction in PC and PC xerogels. The spectra pattern was also compared with its SNCs analogues of xerogels. According to Figure 3, the stacked FTIR spectra of all the compounds showed a prominent peak of alkenyl C-H stretching of a cholesterol group was found at around 2950–2850 cm^-1^. The Alkenyl C=C stretch of cholesterol group was observed at 1605 cm^-1^. The pyridine group was attached to the cholesterol moiety by amide (–CONH) linkage. The C=O (amide) stretching vibrations was assigned at 1690–1650 cm^-1^. Aromatic C=C bending vibrations of the pyridine ring were observed at 1400–1550 cm^-1^. In addition, the additional peaks at 1605 and 1365 cm^−1^ are due to C=N and C-N stretching vibrations of pyridine rings. However, in case of SNC-PC-X1 and SNC-PC-X2 xerogels, the corresponding peaks showed higher transmittance due to the strong affinity of SNPs towards the polar group (pyridine).

### 4.3. FESEM and XRD Analysis

To know the texture of the xerogels, FESEM analysis was performed, and the shape and size of the SNPs inside the xerogels were also analyzed. Figure 4a and b show characteristic nano-wire and nano-fibrous patterns for PC-X1 and PC-X2, respectively. Nano-wire patterns for PC-X1 seem to be loosely constrained to one another, while the nano-fibrous pattern of PC-X2 was closely packed. This eventually controls the thermal stability of the prepared xerogels and their silver analogues. The PC-X1 exhibits typical nano-wire structures with 110–160 nm thickness. However, the uneven close-packing distribution of nano-fibrous structure for PC-X2 exhibits a fiber strand thickness of 70–150 nm. It is clear from Figure 4c and d that SNPs are uniformly anchored in the xerogels (SNC-PC-X1 and SNC-PC-X2) with a predominantly spherical shape, with average particle sizes of 30 nm and 80 nm, respectively. 

Since the crystalline SNPs are developed inside the xerogels, the destruction of the well-patterned fibrous structure of PC was lost, leading to stabilized SNPs. In the case of SNC-PC-X1, the average size of the SNPs is found to be less than 40 nm, whereas the average size of SNPs in SNC-PC-X2 are in the range of 60–80 nm. In addition, the EDX spectrum is provided in the Appendix A, which further evidences the individual elements. Elemental composition analysis by EDX, presented in Appendix A, shows the strongest signal near to 3 keV for xerogel SNCs, which is the typical absorption pattern of a metallic nanocrystalline silver surface. In addition, an X-ray Photoelectron Spectroscopy (XPS) study was performed to evidently clarify the oxidation state of Ag (0 vs. +1) in xerogel SNCs (see Appendix A). 

The crystalline nature of PC and xerogel SNCs was studied using the XRD technique, and the diffractogram was shown in Figure 5. The intense and defined diffraction peaks were observed for PC between 12 and 28°, which are characteristic of a crystalline structure [44,45]. However, sharp and intense Bragg reflections can be seen in case of SNC-PC-X1 and SNC-PC-X2 at 2θ values of 38.2°, 44.2°, 64.4°, 77.5°and 81.6°, which are assigned to the (111), (200), (220), (311) and (222) planes of FCC of silver, respectively [40,41,42]. The diffraction peaks are considerably broad, which indicates that the crystallite size is very small. The XRD pattern shows that formed SNPs are crystalline. 

### 4.4. Thermal Properties

Thermal Gravimetric Analysis (TGA) of PC, PC-X1 and PC-X2 are shown in Figure 6a, which mainly represents weight loss curves (TG). Figure 6b represents weight loss curves of SNC-PC-X1 and SNC-PC-X2. From the TGA plot Figure 6a, we can observe that the TG curve of PC shows an initial decrease in weight, while the temperature approaches 190–192 °C. The PC experiences almost complete degradation in the temperature range of 200–375 °C, where the weight decreases from 99.0% to 0.90%, and the derivate weight loss rate achieved the maximum value at a temperature of 352.5 °C. The xerogels PC-X1 and PC-X2 experience a similar trend with maximum derivate weight loss values at the temperatures of 345.2 °C and 352.6 °C, respectively. The thermal decomposition of SNC-PC-X1 and SNC-PC-X2 appeared in the temperature range 200–350 °C. Two-step degradation was noticed in case of SNCs analogues of xerogels. The first step degradation for SNC-PC-X1 was started from 195 °C to 269.5 °C, with a maximum derivate weight loss value at the temperature of 257.4 °C. The second step degradation was initiated from 270–340 °C, with the maximum derivate weight loss value at the temperature of 326.4 °C. Similarly, a marginal difference was noticed in the case of a SNC-PC-X2 degradation pattern. The first step degradation for SNC-PC-X2 was started from 195 °C to 264.9 °C, with the maximum derivate weight loss value at the temperature 252.1 °C, and the second step degradation in the temperature range of 265–340 °C, with a maximum derivate weight loss value at the temperature of 310.7 °C. The residual mass of SNC-PC-X1 and SNC-PC-X2 at 600 °C were 17.3% and 18.1%, respectively. 

Kinetic and thermodynamic parameters were calculated using Broido’s method [46]. Broido has developed a model, and the activation energy associated with each stage of decomposition was evaluated by this method [46]. Plots of −ln(ln(−1/*y*)) versus 1/*T* (Figure 7a–c) were developed for the decomposition segments of PC, PC-X1, PC-X2, SNC-PC-X1 and SNC-PC-X2. From the plots, the activation energy (*Ea*) and frequency factor (ln*A*) were evaluated. The enthalpy (Δ*H*), entropy (Δ*S*) and free energy (Δ*G*) have also been calculated using standard equations, and are summarized in Table 1, Table 2 and Table 3.

From Table 1, indicated activation energies for PC and PC-X2 were slightly higher when compared to PC, signposting that the decomposition step is considerably faster than PC-X1. The packing in the xerogel state will be reflected by the xerogel properties. If we closely observe the packing arrangement of a PC gelator in PC-X1 and PC-X2, the somewhat loosely confined nanowire structures are dominant in PC-X1. In case of PC-X2, the highly twisted close packing of nanofibers prevail to form the unique structure leading to the slightly higher thermal stability of PC-X2. The SNCs of xerogels (SNC-PC-X1 and SNC-PC-X2) suffer two-step degradation. The results of activation energy from Table 2 and Table 3 indicate that in the decomposition range 200–270 °C, SNC-PC-X1 undergoes faster degradation when compared to SNC-PC-X2, which is associated with the structural stability of gelator PC. The second step degradation was in the range of 270–340 °C, and the activation energies are almost similar in both SNCs xerogels—this was attributed to the stability of capped SNPs with the PC gelator.

### 4.5. Catalytic Properties

It is essential to remove the toxic water pollutant 4-nitrophenol (4-NP) from industrial wastewater for domestic comestible purposes. Therefore, many researchers developed numerous processes to treat the wastewater containing 4-NP with an adsorption technique, microbial degradation, microwave assisted catalytic oxidation, photocatalytic oxidation and electrocatalytic techniques [47,48,49]. The developed silver-induced PC xerogels showed remarkable catalytic activity in the reduction of 4-NP to 4-aminophenol (4-AP) in the presence of aqueous sodium borohydride (NaBH_4_). In order to evaluate the catalytic activity of SNC-PC-X1 and SNC-PC-X2, a model reaction was set and optimized for the concentrations and quantity of the catalyst. Typically, 50 mM aqueous solution of NaBH_4_ was added to 0.1 mM aqueous solution of 4-NP in the presence of 0.05 g of SNC-PC-X1 or SNC-PC-X2, and the intense yellow color of intermediate phenolate ion of 4-NP started to diminish. Within 10 or 15 min, the yellow reaction was turned colorless, indicating the successful conversion of 4-NP to 4-AP. The reaction was carefully monitored by a time-dependent UV-Vis spectroscopic technique. 

The UV-Vis spectrum of 4-NP shows peak at 400 nm (λ_max_), in the presence of catalyst SNC-PC-X1 or SNC-PC-X2, an additional peak appeared at λ_max_ = 300 nm, which is due to the presence of 4-AP (reduction of 4-NP affords 4-AP in the reaction mixture). The maximum conversion of 4-NP to 4-AP was observed after 10 or 15 min, depending on the nanocatalyst used. The catalytic conversion of 4-NP to 4-AP in the presence of NaBH_4_ as a standard reaction for SNC-PC-X1 and SNC-PC-X2, was depicted in Figure 8a and b, respectively. The apparent rate constant (k_app_) and correlation constants (R^2^) for SNC-PC-X1 and SNC-PC-X2 were calculated using linear plots of –ln(A_t_/A_o_) versus reduction time (t), shown in Figure 8c and d, respectively. The percentage conversion of 4-NP to 4-AP with respect to reduction time (t) by SNC-PC-X1 and SNC-PC-X2 nanocatalysts were reported in Table 4 and Table 5, respectively. The apparent rate constant (k_app_) calculated for the catalytic reaction was 6.120 × 10^-3^ sec^-1^ for SNC-PC-X1 and 3.758 × 10^-3^ sec^-1^ for SNC-PC-X2. The correlation coefficient for reduction reaction using of SNC-PC-X1 was found to be 0.887, whereas for SNC-PC-X2 it was found to be 0.927. From Figure 8, it is clear that the reduction reaction follows the pseudo first-order reaction, since the concentration of NaBH_4_ is in large excess when compared to 4-NP. The reaction kinetics can be labeled by the standard equation ln(C/C_0_) = kt, where k is the apparent first-order rate constant (sec^-1^), t is the reaction time, C is the concentrations of 4-NP at time t and C_0_ is the initial concentration of 4-NP.

The recyclability of the catalysts were checked by recovering the catalysts from the reaction mixture, and washed several times with HPLC-grade water and dried at an ambient temperature. The recovered catalyst was used four successive times for the catalytic reduction with almost no loss of catalytic activity, indicating the high stability and durability of the catalysts under normal operating conditions. As shown in Table 6, the recycled catalyst was run for four successive cycles for the catalytic reduction, with a conversion efficiency of around 82%. Moreover, the rate constant of the catalytic reaction was marginally reduced with the increase of the cycle, indicating the stability and recyclability of the nanocatalysts SNC-PC-X1 and SNC-PC-X2.

The metal nanocomposites (palladium, platinum and silver) of dendritic polymers, especially poly(amidoamine) and poly(propyleneimine) dendrimenrs, show k_app_ values of 5.9 × 10^-4^ sec^-1^ and 1.22 × 10^-3^ sec^-1^, respectively [50]. As reported by Chang et al., the k_app_ value reported for SNPs covered with polypyrrole was found to be 1.1 × 10^-3^ s^-1^ [51]. The catalysts made of SNPs covered with functional polystyrene with polyvinylimidazole shows a k_app_ of 5.12 × 10^-4^ s^-1^ [52]. The apparent rate constant (k_app_) reported here is 6.120 × 10^-3^ sec^-1^ and 3.758 × 10^-3^ sec^-1^, which is comparatively superior than the hydrophilic anchored hetrocyclic copolymer micelles with a k_app_ value of 2.12 × 10^-3^ s^-1^ [53]. 

## 5. Conclusions

In summary, two PC xerogels were delicately synthesized along with its SNCs, using an ecofriendly reducing agent, trisodium citrate dehydrate. All the synthesized compounds were spectroscopically characterized. The xerogels composed of SNPs, especially SNC-PC-X1, exhibited a very high catalytic performance in the conversion of 4-NP to 4-AP in the presence of aqueous sodium borohydride at ambient temperatures. The apparent first-order rate constant (k_app_) value for the reduction reactions using SNC-PC-X1 and SNC-PC-X2 was found to be 6.120 × 10^-3^ sec^-1^ and 3.758 × 10^-3^ sec^-1^, respectively. Both the SNCs nanocatalyst xerogels showed improved catalytic performances, with a recyclable conversion efficiency of around 82% after the consecutive fourth run. These findings are important not only for the fabrication of hybrid nanocatalysts composed of supramolecules and metal/metal oxides, but also for the development of advanced catalytic materials with a high catalytic performance, which may provide huge potential for applications in industrial catalysis.

## Figures and Tables

**Figure 1 materials-13-01486-f001:**
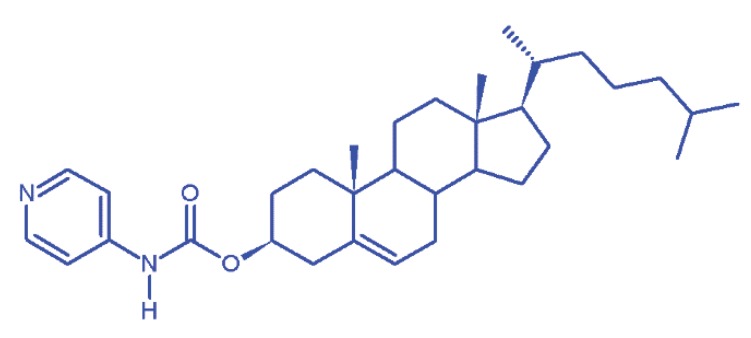
Molecular structure of 4-pyridyl cholesterol (PC).

**Figure 2 materials-13-01486-f002:**
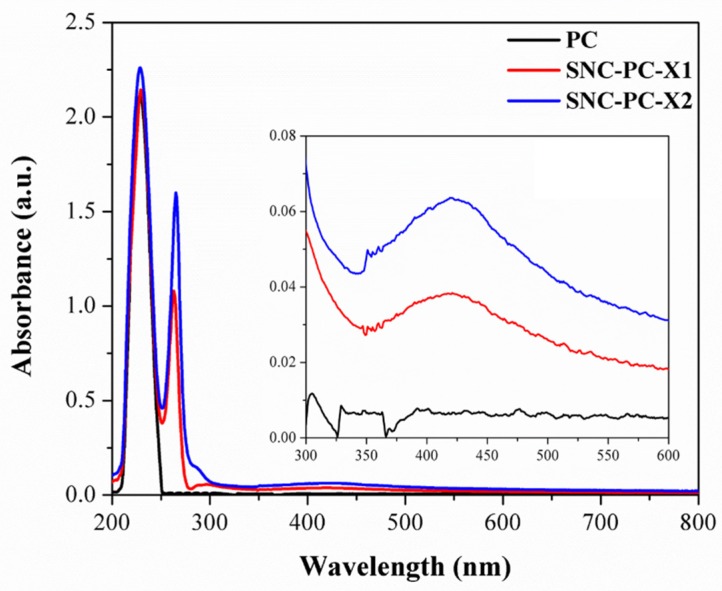
UV-Vis spectrum of 2-weight % solutions of PC, SNC-PC-X1 and SNC-PC-X2 in Tetrahydrofuran (THF) solvent.

**Figure 3 materials-13-01486-f003:**
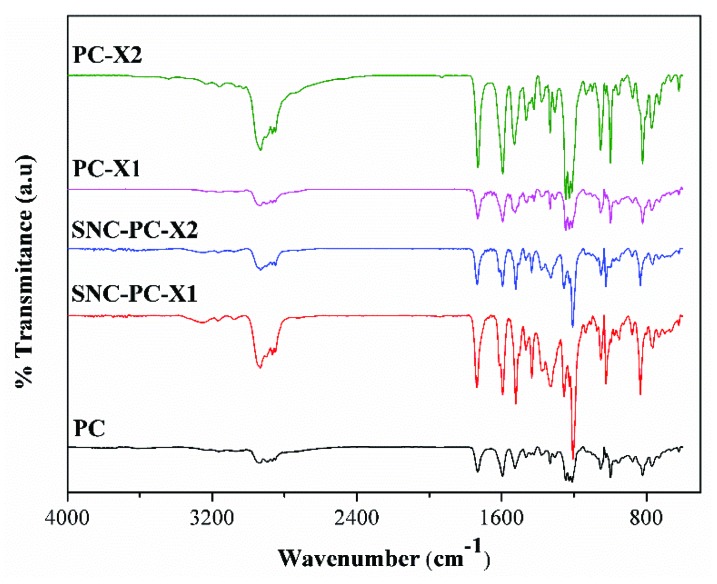
FTIR spectra of PC, PC-X1, PC-X2, SNC-PC-X1 and SNC-PC-X2.

**Figure 4 materials-13-01486-f004:**
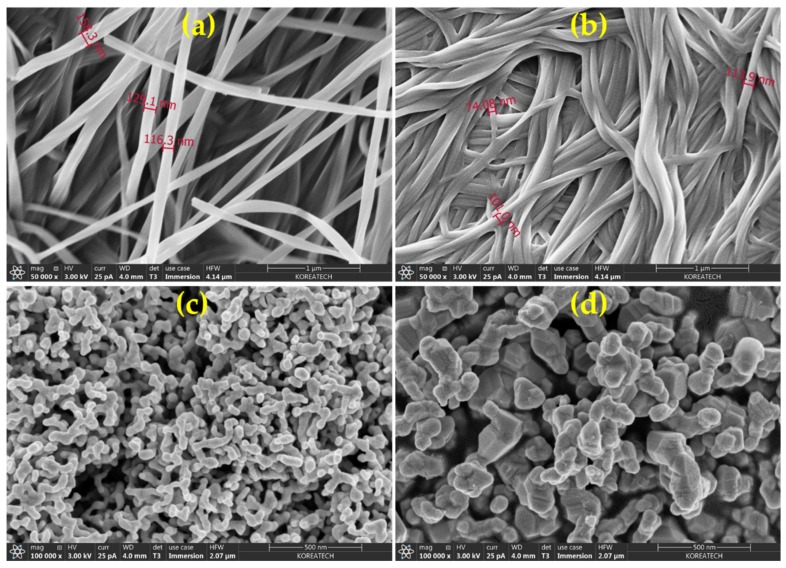
FESEM micrographs of (**a** & **b**) PC-X1 and PC-X2, respectively (Scale = 1 µm); (**c** & **d**) SNC-PC-X1 and SNC-PC-X2, respectively (Scale = 500 nm).

**Figure 5 materials-13-01486-f005:**
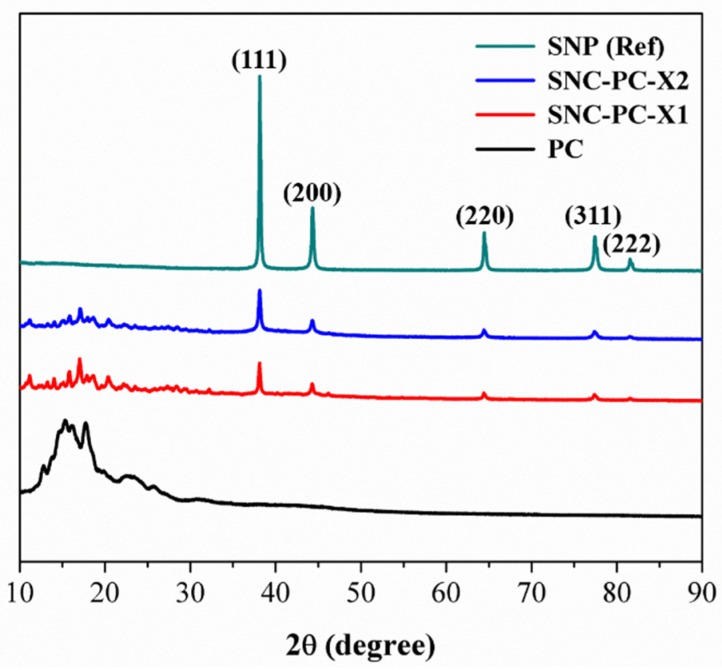
XRD diffractogram of PC, SNC-PC-X1 and SNC-PC-X2 with reference SNPs.

**Figure 6 materials-13-01486-f006:**
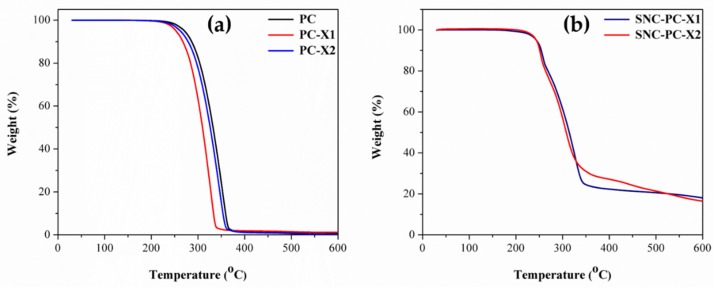
(**a**) TG plot of PC and xerogels (PC-X1 and PC-X2); (**b**) TG plot of xerogel silver nanocomposites (SNC-PC-X1 and SNC-PC-X2), under N_2_ gas at a heating rate of 10 °C/min.

**Figure 7 materials-13-01486-f007:**
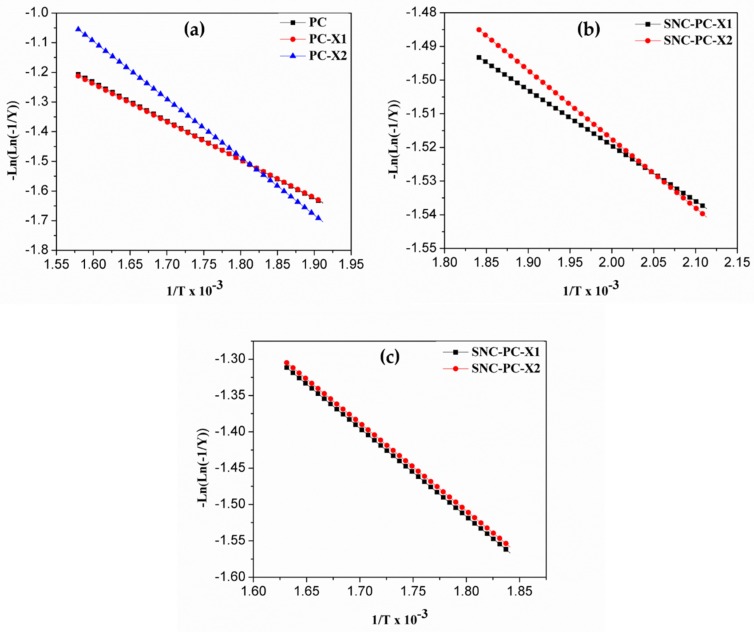
Plots of –Ln(ln(-1/y)) vs 1/T×10^-3^ (**a**) for the decomposition step of PC, PC-X1 and PC-X2 in the range 250–360 °C; (**b**) for the first decomposition step of SNC-PC-X1 and SNC-PC-X2 in the range 200–270 °C; and (**c**) for the second decomposition step of SNC-PC-X1 and SNC-PC-X2 in the range 270–340 °C.

**Figure 8 materials-13-01486-f008:**
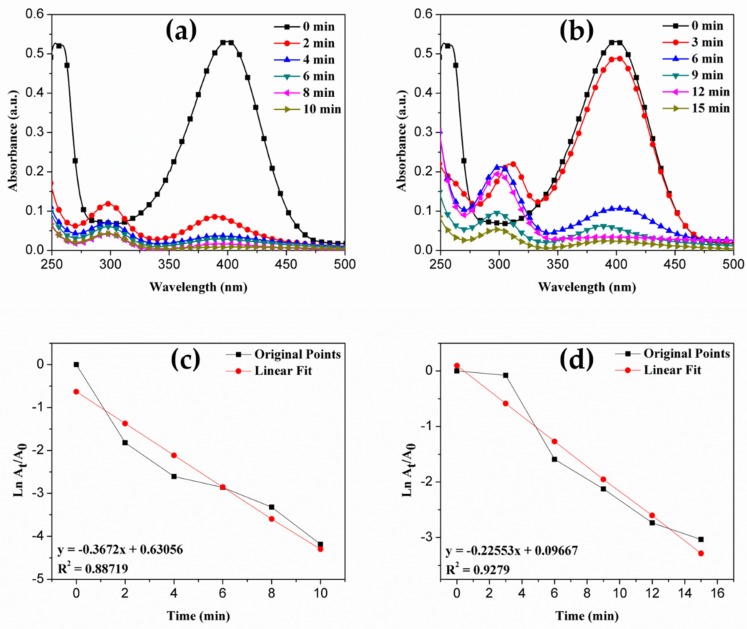
Time-dependent UV-Vis spectra representing the catalytic reduction of 4-NP into 4-AP using (**a** & **b**) nanocatalyst SNC-PC-X1 and SNC-PC-X2, respectively; and linear plot of –ln(At/Ao) versus reduction time for (**c** & **d**) nanocatalyst SNC-PC-X1 and SNC-PC-X2, respectively.

**Table 1 materials-13-01486-t001:** Kinetic and thermodynamic parameters of PC, PC-X1 and PC-X2 at the decomposition range of 250–360 °C.

Compound	E_a_ (kJ/mol) × 10^-3^	ln*A*	∆H (kJ/mol)	∆S (kJ/K)	∆G (kJ/mol)
PC	26.045	−5.542	−4.780	−162.464	93.900
PC-X1	24.533	−5.599	−4.781	−162.479	97.172
PC-X2	25.389	−4.269	−4.768	−162.118	93.699

**Table 2 materials-13-01486-t002:** Kinetic and thermodynamic parameters of SNC-PC-X1 and SNC-PC-X2 at the decomposition range of 200–270 °C.

Compound	E_a_ (kJ/mol) × 10^-3^	ln*A*	∆H (kJ/mol)	∆S (kJ/K)	∆G (kJ/mol)
SNC-PC-X1	3.157	−9.205	−4.222	−164.186	83.444
SNC-PC-X2	3.913	−8.943	−4.221	−164.166	83.435

**Table 3 materials-13-01486-t003:** Kinetic and thermodynamic parameters of SNC-PC-X1 and SNC-PC-X2 at the decomposition range of 270–340 °C.

Compound	E_a_ (kJ/mol) × 10^-3^	ln*A*	∆H (kJ/mol)	∆S (kJ/K)	∆G (kJ/mol)
SNC-PC-X1	23.260	−5.717	−4.782	−161.894	93.570
SNC-PC-X2	23.160	−5.714	−4.782	−161.527	93.358

**Table 4 materials-13-01486-t004:** Catalytic reduction of 4-NP to 4-AP using SNC-PC-X1 nanocatalyst.

Time (min)	Percentage Conversion of 4-NP to 4-AP
0	0
2	83.80
4	92.61
6	94.29
8	96.38
10	98.48

**Table 5 materials-13-01486-t005:** Catalytic reduction of 4-NP to 4-AP using SNC-PC-X2 nanocatalyst.

Time (min)	Percentage Conversion of 4-NP to 4-AP
0	0
3	7.54
6	79.67
9	88.06
12	93.51
15	95.19

**Table 6 materials-13-01486-t006:** Recyclable efficiency of nanocatalysts.

Recycle Attempts	Catalyst SNC-PC-X1	Catalyst SNC-PC-X2
1	95.21	91.62
2	93.15	87.49
3	86.13	85.69
4	82.48	81.93

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
