# Peer review of "Silver-Nanoparticles Embedded Pyridine-Cholesterol Xerogels as Highly Efficient Catalysts for 4-nitrophenol Reduction"

_materials, 2020, doi:10.3390/ma13071486_

Round 1

Reviewer 1 Report

This work reported the preparation of two xerogel material for catalytic reduction of 4- nitrophenol. The study was well conducted and the catalyst is highly active and recylable. Therefore, I recommend accept this work with a minor revision that the authors should discuss the reason for differences of the catalytic performance for these two materials to provide some guidance for catalyst design.

Author Response

Journal: Materials (ISSN 1996-1944)

Title: Silver-nanoparticles embedded Pyridine-Cholesterol Xerogels as Highly Efficient Catalysts for 4-nitrophenol Reduction  

Type: Article

Manuscript ID: materials-740921     

Dear Editor,

Thank you for your useful comments and suggestions on the structure of our manuscript. We have modified the manuscript accordingly, and detailed corrections are listed below point by point:

Reviewer # 1

 Open Review

English language and style

( ) Extensive editing of English language and style required
( ) Moderate English changes required
(x) English language and style are fine/minor spell check required
( ) I don't feel qualified to judge about the English language and style

Yes

Can be improved

Must be improved

Not applicable

Does the introduction provide sufficient background and include all relevant references?

( )

(x)

( )

( )

Is the research design appropriate?

(x)

( )

( )

( )

Are the methods adequately described?

(x)

( )

( )

( )

Are the results clearly presented?

(x)

( )

( )

( )

Are the conclusions supported by the results?

(x)

( )

( )

( )

Comments and Suggestions for Authors:

This work reported the preparation of two xerogel material for catalytic reduction of 4- nitrophenol. The study was well conducted and the catalyst is highly active and recylable. Therefore, I recommend accept this work with a minor revision that the authors should discuss the reason for differences of the catalytic performance for these two materials to provide some guidance for catalyst design.

  • Insights on the catalytic performances of the designed catalysts are provided in the modified manuscript and highlighted.
  • Grammatical and typing errors are carefully checked and modified.

Thank you very much for your suggestions and comments on the manuscript materials-740921.                                                                                                                 

Date: 16.03.2020                                                                                            

With Regards

Dr. Ganesh Shimoga                                            

Reviewer 2 Report

The manuscript entitled “Silver-nanoparticles embedded Pyridine-Cholesterol Xerogels as Highly Efficient Catalysts for 4-nitrophenol Reduction” has described the efficient reduction of 4-nitrophenol to 4-aminophenol using two xerogels synthesized from 4-pyridyl cholesterol (PC) and silver-nanocomposites in the presence of aqueous sodium borohydride. Additionally, the apparent first-order rate constant (kapp) value for the reduction reactions using SNC-PC-X1 and SNC-PC-X2 has been found to be 6.120x10-3 sec-1 and 3.758x10-3 sec-1 respectively. The present work is good and could be beneficial to the researchers working in this field. According to this reviewer, this manuscript could be accepted after incorporating the following comments/suggestions.

They are:

  1. Typographical errors should be addressed properly.
  2. Some relevant references are missing. The following references should be discussed and cited in the main body of the manuscript (Coordination Chemistry Reviews 2015, 287, 114-136; ACS sustainable chemistry & engineering 2017, 5 (5), 3637-364; Din, M.I., Khalid, R., Hussain, Z., Hussain, T., Mujahid, A., Najeeb, J. and Izhar, F., 2019. Nanocatalytic Assemblies for Catalytic Reduction of Nitrophenols: A Critical Review. Critical reviews in analytical chemistry, pp.1-17.)

Author Response

Journal: Materials (ISSN 1996-1944)

Title: Silver-nanoparticles embedded Pyridine-Cholesterol Xerogels as Highly Efficient Catalysts for 4-nitrophenol Reduction  

Type: Article

Manuscript ID: materials-740921     

Dear Editor,

Thank you for your useful comments and suggestions on the structure of our manuscript. We have modified the manuscript accordingly, and detailed corrections are listed below point by point:

Reviewer # 2

Open Review

English language and style

( ) Extensive editing of English language and style required
(x) Moderate English changes required
( ) English language and style are fine/minor spell check required
( ) I don't feel qualified to judge about the English language and style

Yes

Can be improved

Must be improved

Not applicable

Does the introduction provide sufficient background and include all relevant references?

( )

(x)

( )

( )

Is the research design appropriate?

(x)

( )

( )

( )

Are the methods adequately described?

(x)

( )

( )

( )

Are the results clearly presented?

(x)

( )

( )

( )

Are the conclusions supported by the results?

(x)

( )

( )

( )

Comments and Suggestions for Authors:

The manuscript entitled “Silver-nanoparticles embedded Pyridine-Cholesterol Xerogels as Highly Efficient Catalysts for 4-nitrophenol Reduction” has described the efficient reduction of 4-nitrophenol to 4-aminophenol using two xerogels synthesized from 4-pyridyl cholesterol (PC) and silver-nanocomposites in the presence of aqueous sodium borohydride. Additionally, the apparent first-order rate constant (kapp) value for the reduction reactions using SNC-PC-X1 and SNC-PC-X2 has been found to be 6.120x10-3 sec-1 and 3.758x10-3 sec-1 respectively. The present work is good and could be beneficial to the researchers working in this field. According to this reviewer, this manuscript could be accepted after incorporating the following comments/suggestions.

They are:

  1. Typographical errors should be addressed properly.
  • Grammatical and typing errors are carefully checked and modified.
  1. Some relevant references are missing. The following references should be discussed and cited in the main body of the manuscript (Coordination Chemistry Reviews 2015, 287, 114-136; ACS sustainable chemistry & engineering 2017, 5 (5), 3637-364; Din, M.I., Khalid, R., Hussain, Z., Hussain, T., Mujahid, A., Najeeb, J. and Izhar, F., 2019. Nanocatalytic Assemblies for Catalytic Reduction of Nitrophenols: A Critical Review. Critical reviews in analytical chemistry, pp.1-17.)
  • References 47, 48 and 49 are included in the modified manuscript and highlighted.
  • Relevant references are now included in the modified manuscript and change was highlighted.

Thank you very much for your suggestions and comments on the manuscript materials-740921.                  

Date: 16.03.2020

With Regards

Dr. Ganesh Shimoga                                                                        

Reviewer 3 Report

The authors describe in their publication entitled “Silver-nanoparticles embedded Pyridine-Cholesterol Xerogels as Highly Efficient Catalysts for 4-nitrophenol Reduction” the synthesis, characterization and testing of two xerogels. The manuscript itself is drafted in a logical way, but the manuscript needs a major revision prior consideration for acceptance in the journal. The authors should pay attention to the following issues which need to be solved:

Materials and Methods:

1) The Synthesis of PC is not fully clear (see section 2.1, lines 96-98): How was the product obtained – by precipitation or by removing the solvent?

2) Analytics for PC are completely missing. Please add melting point (adding literature value for comparison, too), and elemental analysis.

3) Basic analytics for the PC xerogels are missing: What is the final weight of the formed xerogels after vacuum drying? Please add also TG/DSC information as well as elemental analysis.

4) Basic analytics for Ag/PC xerogels are missing, too. Please add elemental analysis (including Ag content) as well as TG/DSC information.

Results and discussion:

1) Section 3.3. SEM/XRD: From Figure 4 it is obvious that the fibrous PC structure is destroyed after Ag loading. Furthermore, it is obvious from Fig. 4 c and d that the particles exhibit crystallite planes. The authors must clearly address this totally different behavior and must explain the different morphologies – not only the doubled size in c and d – but also the destruction of the PC primary structure by comparison with analytical data (see above). It is completely unknown why this behavior (destruction of xerogel structure) is present after Ag loading and the authors must clarify this precisely. Furthermore, the authors need to get someone familiar with discussing XRD data: Especially lines 214-215 are totally unscientific and not true. Furthermore, the authors must give information about the size of Ag crystallites and present additional SEM/EDX spectra as well as mapping information.

2) In addition, the authors must perform additional XPS measurements to clarify the oxidation state of Ag (0 vs. +1) unambiguously.

3) The authors must give information on comparison data from their catalytic measurements. There is no single information about the performance of the as-synthesized materials compared to literature values.

Author Response

Journal: Materials (ISSN 1996-1944)

Title: Silver-nanoparticles embedded Pyridine-Cholesterol Xerogels as Highly Efficient Catalysts for 4-nitrophenol Reduction  

Type: Article

Manuscript ID: materials-740921     

Dear Editor,

Thank you for your useful comments and suggestions on the structure of our manuscript. We have modified the manuscript accordingly, and detailed corrections are listed below point by point:

Reviewer # 3

Open Review

English language and style

( ) Extensive editing of English language and style required
( ) Moderate English changes required
(x) English language and style are fine/minor spell check required
( ) I don't feel qualified to judge about the English language and style

Yes

Can be improved

Must be improved

Not applicable

Does the introduction provide sufficient background and include all relevant references?

( )

(x)

( )

( )

Is the research design appropriate?

( )

( )

(x)

( )

Are the methods adequately described?

( )

( )

(x)

( )

Are the results clearly presented?

( )

( )

(x)

( )

Are the conclusions supported by the results?

( )

( )

(x)

( )

Comments and Suggestions for Authors:

The authors describe in their publication entitled “Silver-nanoparticles embedded Pyridine-Cholesterol Xerogels as Highly Efficient Catalysts for 4-nitrophenol Reduction” the synthesis, characterization and testing of two xerogels. The manuscript itself is drafted in a logical way, but the manuscript needs a major revision prior consideration for acceptance in the journal. The authors should pay attention to the following issues which need to be solved:

Materials and Methods:

1. The Synthesis of PC is not fully clear (see section 2.1, lines 96-98): How was the product obtained – by precipitation or by removing the solvent?

  • PC is obtained by precipitation method and the purification is done by repeated precipitation and removing solvents by rotary evaporator.
  • The sentences in Section 2.1, lines 96-98 is modified and highlighted in the manuscript.

2. Analytics for PC are completely missing. Please add melting point (adding literature value for comparison, too), and elemental analysis.

  • The melting point of PC is added with elemental analysis details and highlighted.
  • The literature [34] is quoted for justification.

3. Basic analytics for the PC xerogels are missing: What is the final weight of the formed xerogels after vacuum drying? Please add also TG/DSC information as well as elemental analysis.

  • The elemental analysis is provided and highlighted.
  • TGA data with kinetic parameters are given in Section 3.4 Thermal Properties.

4. Basic analytics for Ag/PC xerogels are missing, too. Please add elemental analysis (including Ag content) as well as TG/DSC information.

  • The elemental analysis is provided and highlighted.
  • TGA data with kinetic parameters are given in Section 3.4 Thermal Properties.
  • Theoretical Ag content in the xerogels were mentioned.

Results and discussion: 

1. Section 3.3. SEM/XRD: From Figure 4 it is obvious that the fibrous PC structure is destroyed after Ag loading. Furthermore, it is obvious from Fig. 4 c and d that the particles exhibit crystallite planes. The authors must clearly address this totally different behavior and must explain the different morphologies – not only the doubled size in c and d – but also the destruction of the PC primary structure by comparison with analytical data (see above). It is completely unknown why this behavior (destruction of xerogel structure) is present after Ag loading and the authors must clarify this precisely. Furthermore, the authors need to get someone familiar with discussing XRD data: Especially lines 214-215 are totally unscientific and not true. Furthermore, the authors must give information about the size of Ag crystallites and present additional SEM/EDX spectra as well as mapping information.

  • The line 214-215 is removed from the manuscript.
  • The EDX plot is provided in supplementary information document.
  • A small paragraph is included in the manuscript, which explains the destruction of fibrous pattern of PC.

2. In addition, the authors must perform additional XPS measurements to clarify the oxidation state of Ag (0 vs. +1) unambiguously.

  • The XPS data is provided in supporting information.

3. The authors must give information on comparison data from their catalytic measurements. There is no single information about the performance of the as-synthesized materials compared to literature values.

  • Comparison of the apparent rate constant (kapp) obtained from our experiment with literature values are described in a short paragraph. Also, relevant references were added for justification.

Thank you very much for your suggestions and comments on the manuscript materials-740921.                                                                                                             

Date: 16.03.2020                                                                                                 

With Regards

Dr. Ganesh Shimoga                                                                        

Round 2

Reviewer 3 Report

The authors reworked their manuscript as needed from my suggestions. Unfortunately, the Ag content in the materials is still undetermined. Giving theoretical values is insufficient. The authors must included actual Ag content from AAS or ICP-OES measurements and must correct the values given in section 2.3 in a final minor revision.

Author Response

Journal: Materials (ISSN 1996-1944)

Title: Silver-nanoparticles embedded Pyridine-Cholesterol Xerogels as Highly Efficient Catalysts for 4-nitrophenol Reduction  

Type: Article

Manuscript ID: materials-740921     

Dear Editor,

Thank you for your useful comments and suggestions on the structure of our manuscript. We have modified the manuscript accordingly, and detailed corrections are listed below point by point:

Reviewer # 3

Open Review 

English language and style

( ) Extensive editing of English language and style required
( ) Moderate English changes required
(x) English language and style are fine/minor spell check required
( ) I don't feel qualified to judge about the English language and style

Yes

Can be improved

Must be improved

Not applicable

Does the introduction provide sufficient background and include all relevant references?

( )

(x)

( )

( )

Is the research design appropriate?

( )

(x)

( )

( )

Are the methods adequately described?

(x)

( )

( )

( )

Are the results clearly presented?

( )

(x)

( )

( )

Are the conclusions supported by the results?

( )

(x)

( )

( )

Comments and Suggestions for Authors:

The authors reworked their manuscript as needed from my suggestions. Unfortunately, the Ag content in the materials is still undetermined. Giving theoretical values is insufficient. The authors must included actual Ag content from AAS or ICP-OES measurements and must correct the values given in section 2.3 in a final minor revision.

  • The experimental results of Ag content by ICP-OES measurement is now presented with relative standard deviation (RSD) values in the section 2.3.

Thank you very much for your suggestions and comments on the manuscript materials-740921.                                                                                                                    

Date: 19.03.2020

With Regards

Dr. Ganesh Shimoga                                                                  
